# Oxygenation before Endoscopic Sedation Reduces the Hypoxic Event during Endoscopy in Elderly Patients: A Randomized Controlled Trial

**DOI:** 10.3390/jcm9103282

**Published:** 2020-10-13

**Authors:** Hyunil Kim, Jeong Nam Hyun, Kyong Joo Lee, Hyun-Soo Kim, Hong Jun Park

**Affiliations:** Division of Gastroenterology and Hepatology, Department of Internal Medicine, Yonsei University Wonju College of Medicine, Ilsan-ro 20, Wonju 26426, Korea; kimhyunil@empas.com (H.K.); charart@naver.com (J.N.H.); smild123@yonsei.ac.kr (K.J.L.); hyskim@yonsei.ac.kr (H.-S.K.)

**Keywords:** sedation, sedative endoscopy, oxygen, pre-oxygenation

## Abstract

Background: Sedation endoscopy increases patient and examiner satisfaction but involves complications. The most serious complication is hypoxia, the risk factors for which are old age, obesity, and American Society of Anesthesiologists physical status of 3 or greater. However, clear evidence of oxygenation during sedation endoscopy for elderly people is lacking in US, European, and Korean guidelines. Method: This study was conducted for 1 year starting in August 2018 to evaluate whether pre-oxygenation use 1 min before sedation endoscopy could reduce the incidence of hypoxia in patients older than 65 years of age. A total of 70 patients were divided into the non-oxygenated group (*n* = 35; control group) and oxygen-treated group (*n* = 35; experimental group) during endoscopy. Result: The incidence of hypoxia was 28 (80%) in the control group versus 0 (0%) in the pre-oxygenated group. Factors related to hypoxia in the non-oxygenated group were a relatively high dose of midazolam and concomitant injection with narcotic analgesics such as pethidine. Conclusion: The incidence of hypoxia during sedation endoscopy is high in patients over 65 years, but oxygenation during endoscopic sedation in elderly people can significantly reduce the incidence of intraprocedural hypoxic events.

## 1. Introduction

Sedation can suppress patient consciousness and cause anterograde amnesia during endoscopy as well as decrease patient anxiety and pain at the time of examination [1]. This improves patient satisfaction with endoscopy and increases their willingness to undergo the next examination [2,3]. However, unintentional deep sedation can lead to various complications, including hypoxia, hypotension, and arrhythmia [2,4,5]. In particular, the frequency of hypoxia related to endoscopic sedation is reportedly 6–18% depending on the drug and dosage; furthermore, hypoxia associated with hypoventilation accounts for up to 66% of cases [6]. Major risk factors associated with sedation-related hypoxia include older age, obesity, and American Society of Anesthesiologists (ASA) physical status [7].

Chronic hypoxemia and hypercapnia due to chronic ventilation-perfusion mismatch are often comorbid, especially in elderly patients whose ability to cope with hypoxia caused by sedation may be impaired [8]. An analysis of 13 million endoscopies performed in the United States over nine years revealed that the incidence of complications from gastrointestinal endoscopy increases with age [9].

Therefore, the US and European guidelines for sedation and anesthesia in gastrointestinal endoscopy recommend oxygenation during sedation [7,10]. However, there are no stipulated guidelines in Korea. In particular, the anesthesiologists recommend pre-oxygenation, in which oxygen is given before sedation [11,12,13]. 

Nevertheless, there are few studies on the efficacy of pre-oxygenation in older patients during endoscopy. Therefore, this prospective comparative study aimed to investigate the ability of pre-oxygenation to reduce the frequency of hypoxia during endoscopy in elderly patients.

## 2. Materials and Methods

### 2.1. Study Overview

This study was approved by the institutional review board of Wonju Severance Christian Hospital (approval no. CR318066) and submitted to the Clinical Research Information Service at the Centers for Disease Control and Prevention (approval no. KCT0003693).

This single-center prospective randomized study was conducted in the endoscopy room of Wonju Severance Christian Hospital, Yonsei University. Patients submitted informed consent prior to participating in the clinical study. Investigator randomly assigned the patients to the experimental or control group by the sequential sealed opaque envelope method on the test day. The endoscopic oxygenation study was performed non-blinded because it was difficult to proceed blinded.

The primary purpose of this study was to compare the hypoxia incidence during endoscopy under sedation between the pre-oxygenated and non-oxygenated groups. The secondary purpose was to determine the differences in demographic factors between the hypoxia and non-hypoxia groups and compare the underlying disease and endoscopy-related factors (drug types and doses, endoscopic procedures, and endoscopic procedure time). Oxygen saturation tracings were taken during endoscopic sedation.

### 2.2. Subjects

The subjects enrolled in this study were elderly patients who were scheduled to undergo sedation endoscopy. According to the World Health Organization definition, an elderly individual is 65 years or older. Patients at high risk of hypoxia with severe underlying diseases (ASA score of 3 or higher) were excluded.

This study was a superiority test, and the number of study subjects was obtained through the following process. In the reference study, the hypoxic event in the oxygenated and non-oxygenated groups was 0% and 23.3%, respectively. However, it was possible that more patients were classified as having hypoxia in the reference study by setting the definition of hypoxia as an oxygen saturation below 92%, which is higher than 90% and generally accepted as hypoxia [14]. In addition, the reference study included young and older patients (mean age of the oxygenated group: 51.1 years; mean age of the non-oxygenated group: 46.6 years); thus, fewer patients would have experienced hypoxia. Accordingly, we set the effect size as 30% of the expected difference between the two groups.

(1)Type of research: Superiority test (Fisher’s exact test)(2)Alpha: 0.05; Beta: 0.20; Power: 0.80; Two-tailed test; Allocation ratio N2/N1: 1(3)Proportion p1: (0%), proportion p2: 0.3 (30%)

Using the above figures, a total of 33 patients would be required in each group, and a total of 70 patients were needed with a dropout rate of 5% (four patients).

### 2.3. Study Process and Measurement Variables

(1)Collect patient characteristics such as age, sex, underlying disease, medication history, and previous history of endoscopic sedation and measure their vital signs and oxygen saturation before sedation.(2)Administer oxygen at 2 L/min via nasal prongs 1 min before endoscopy in the pre-oxygenation group.(3)Inject midazolam or propofol (+/− pethidine) at 1–2-min intervals until moderate sedation is achieved.(4)Check oxygen saturation every 1 min during endoscopic sedation. In case of hypoxia below 90% oxygen saturation, follow the procedures below.(5)Check vital signs such as consciousness level, blood pressure, and oxygen saturation until the end of the endoscopic procedure.

** Management process in patients with hypoxia ** 

① Raise oxygen supplement from 2 to 6 L/min in the pre-oxygenated group and immediately start oxygen at a rate of 2 L/min via nasal prongs in the non-oxygenated group. 

② To induce arousal, call the patient’s name or provide a stimulus such as shaking or pain. 

③ If hypoxia persists even after oxygen administration and stimulation, intravenously administer 0.3 mg of flumazenil (midazolam antagonist) for the midazolam group and naloxone for the pethidine group at the physician’s discretion. 

④ If hypoxia persists after the above procedures, first aid should be performed in the order of airway maneuver, air mask bag unit (AMBU) bagging, and intubation.

### 2.4. Statistics

A statistical significance analysis was performed using SPSS version 20.0 (SPSS Inc., Chicago, IL, USA). Continuous variables were analyzed using an independent sample t-test or one-way analysis of variance, while categorical variables were analyzed using the χ^2^ test. Values of *p* < 0.05 were considered statistically significant.

## 3. Results

This study was performed for 1 year starting in August 2018. The mean patient age was 73.8 ± 6.2 years; 28 (40%) patients were men and 42 (60%) patients were women. All patients underwent upper gastrointestinal endoscopy (esophagogastroduodenoscopy or endoscopic ultrasonography). Among them, 44 patients (63%) had medical histories such as hypertension, diabetes, heart disease, lung disease, cerebrovascular disease, or renal disease, while 29 patients (41.4%) were obese with a body mass index >25. The Mallampati score, a surrogate predictor of obstructive sleep apnea, was 1 or 2 points in 29 (84%) patients and 4 points in four (5.7%) patients.

The mean oxygen saturation before endoscopy was 95.6 ± 2.4%. The mean dosage of midazolam was 2.7 ± 0.6 mg for 68 (97.1%) patients, that of propofol was 13.6 ± 12.0 mg for 44 (62.9%) patients, and that of pethidine was 12.1 ± 21.6 mg for 17 (24.3%) patients.

The mean patient age of the oxygenated group was 75.8 years, significantly higher than that of the non-oxygenated group (71.8 years). 

There was no statistically significant intergroup difference in the degree of obesity, smoking history, alcohol history, medical history, Mallampati score, or sedatives used such as midazolam, propofol, and pethidine that could affect oxygen saturation (Table 1).

There was no intergroup difference in the average oxygen saturation before sedation endoscopy but for one patient with a hypoxic event in the non-oxygenated group. During the procedure, surprisingly, hypoxia occurred in 28 (80%) patients in the non-oxygenated group versus no patient in the oxygenated group (Table 2). Since no hypoxia events occurred in the oxygenation group, we used exact logistic regression on the basis that the odds ratio is estimated to be similar to the relative risk when the outcome ratio is small. As a result, the relative risk of hypoxia in the non-oxygenated group was 13.12 higher than that in the pre-oxygenated group (Table 3).

On a graph of serial oxygen saturation taken every 1 min, the non-oxygenated group showed a rapid decrease in oxygen saturation from 1 min and gradually recovered from 5 min, but the patients in the pre-oxygenated group showed a tendency to maintain oxygen saturation >95% (Figure 1).

When the oxygen saturations of 28 hypoxic patients were plotted at 1-min intervals, the trend of hypoxia events from 1 min to 5 min became clearer (Figure 2 and Figure 3). In an additional sub-analysis created by dividing the non-oxygenated group into two groups (with or without hypoxia), high blood pressure was more common in the group without hypoxia, but the number was small and not considered significant, and we could see that the dosages of midazolam and pethidine were higher in the hypoxic group. There was a trend toward a high Mallampati score in the hypoxia group despite a lack of statistical significance (Table 4).

## 4. Discussion

Although sedative endoscopy is a good way to minimize patient anxiety and increase patient satisfaction with the endoscopic procedure, many studies have shown a risk of related complications [15,16,17,18,19,20,21,22,23].

The most important complication related to endoscopic sedation is hypoxia, and it is well known that the associated risk factors are old age, underlying disease, a high ASA score, obesity, and a high risk of sleep apnea [9]. To prevent hypoxia, the use of oxygen during the procedure is recommended in the US, Europe, and Korea, but there are no clear recommendations of how or in which cases [7,10]. The ASA and the American Society for Gastrointestinal Endoscopy recommend that supplemental oxygen be considered for moderate sedation and be required for all procedures with intended deep sedation. Supplemental oxygen should be administered if hypoxemia is anticipated or develops. The revised European Guideline (2015) states that continuous supplemental oxygen is indicated during non-anesthesia provider-administered propofol for endoscopy. However, the potential benefit of routine prophylactic oxygen supplementation in terms of decreased cardiopulmonary complications is unclear. Both guidelines considered old age a risk factor for sedative endoscopy complications but provide no solutions.

It is reported that oxygenation is provided during sedation endoscopy in 39% of Italian studies, 35–42% of German studies, 20–41% of Greek studies, and 58–78% in Swiss studies. In a Korean survey, only 27% of respondents reported using oxygen during endoscopic sedation [24,25]. These findings show discrepancies between the guidelines and real-life practice.

Therefore, this study presents the usefulness of oxygenation 1 min before endoscopy to effectively prevent hypoxia (the most serious complication of sedation endoscopy) in elderly patients (the most common risk factor of hypoxia) and provide a clear basis for oxygenation during endoscopy sedation.

Anesthesiologists recommend providing oxygenation 1 min before the procedure, when the functional residue is filled with 2500 mL of oxygen and where O_2_-CO_2_ exchange does not actually occur due to pre-oxygenation and it can theoretically last 10 min even under apnea conditions [11,12,13]. However, in other studies, the target population of pre-oxygenation therapy has included not only patients with high risk but also patients with low risk [14].

In this study, the incidence of hypoxia during upper gastrointestinal endoscopy under sedation was over 80% in people older than 65 years. In terms of time, the oxygen saturation rapidly decreased 1 min after the sedative injection when the drug’s effect appeared and lasted around 5 min. Oxygenation delivered 1 min before endoscopic sedation prevented hypoxia, resulting in no cases of hypoxia in that group. Therefore, oxygenation provided 1 min before the procedure may be the best way to prevent hypoxia in patients aged 65 years or older.

The primary limitation of this study was that there was no comparison between oxygenation just before the procedure and oxygenation 1 min before endoscopy. However, from the clinical point of view, oxygenation provided 1 min before the endoscopy is not a hassle, so we think that this will not be difficult to perform in actual clinical practice.

The secondary limitation of this study is that it included a rather small group. A large-scale study including the risk factors related to hypoxia in endoscopic sedation would provide clearer results of the efficacy of pre-oxygenation for endoscopic procedures.

The third limitation was our sedation protocols. We adopted midazolam-based balanced propofol sedation whereby an initial 2–3 mg midazolam was given to patients followed by an additional 10–20 mg propofol until the patient was sedated enough. Exceptionally, we adopted a propofol mono sedation protocol as an alternative regimen for the patients with liver disease, especially liver cirrhosis. In the case of propofol sedation, we used 30 mg propofol as an initial dose and 10–20 mg propofol for additional doses consecutively. So, this effect of preoxygenation to sedation may concern mainly midazolam-based balanced propofol sedation rather than propofol mono sedation. To clarify the effects of preoxygenation on propofol mono sedation, more studies may be needed.

## 5. Conclusions

In conclusion, hypoxia during endoscopic sedation can occur in up to 80% of elderly patients, usually within 1–5 min of the start of the procedure. Therefore, providing oxygenation 1 min before endoscopy can be an effective way to prevent hypoxia.

## Figures and Tables

**Figure 1 jcm-09-03282-f001:**
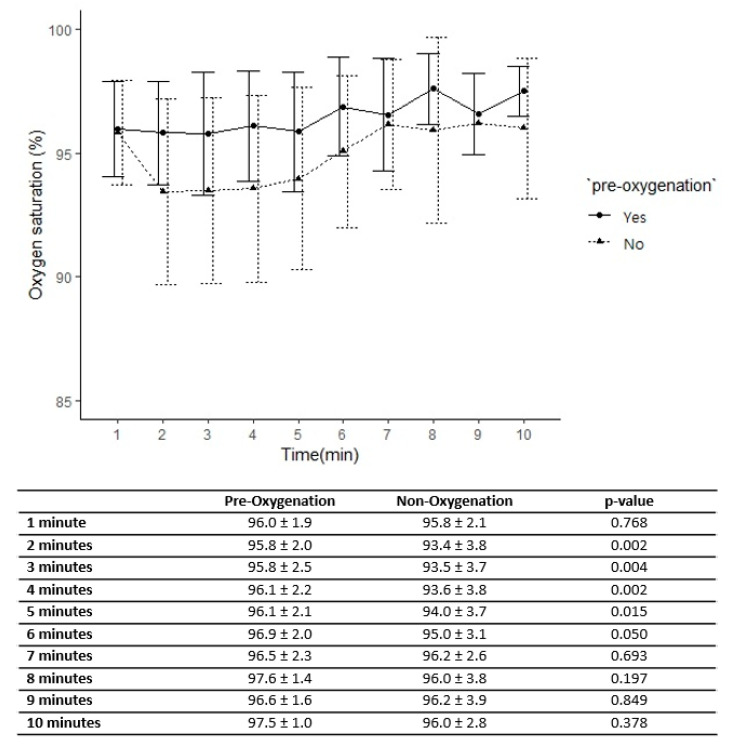
The trend of mean oxygen saturation (%) by times (1 min interval) in both groups.

**Figure 2 jcm-09-03282-f002:**
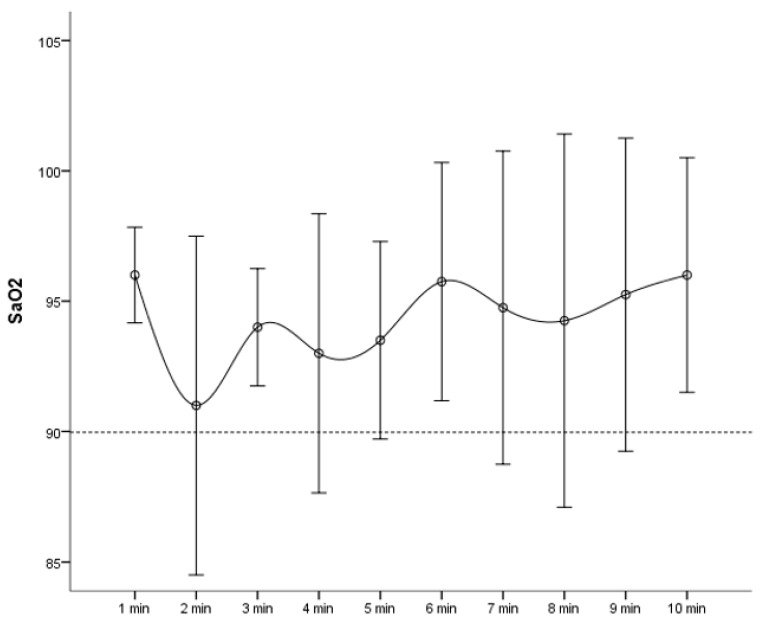
Mean oxygen saturation (%) in patients with hypoxic event by time (minutes) (*n* = 28).

**Figure 3 jcm-09-03282-f003:**
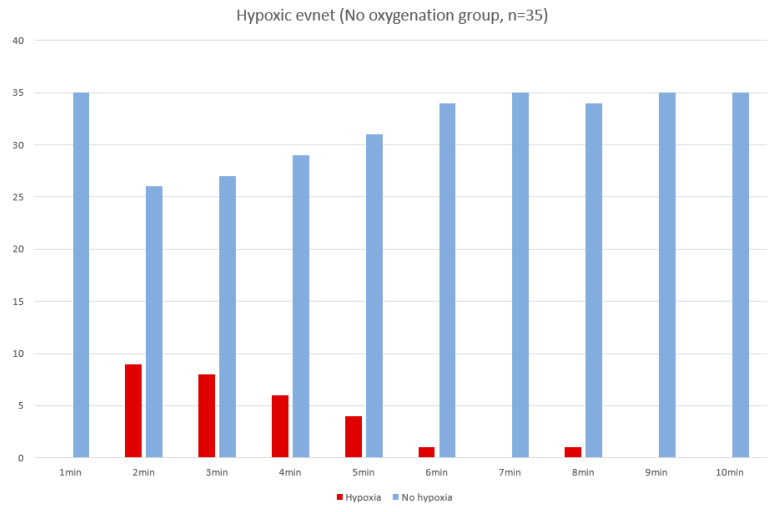
The frequency of hypoxic event (*n*) in non-oxygenation group by time (minutes) (*n* = 35).

**Table 1 jcm-09-03282-t001:** Patients’ baseline characteristics in both groups.

	Oxygenated Group (*n* = 35)	Non-Oxygenated Group (*n* = 35)
Sex (male, %)	12 (34.3%)	16 (45.7%)
Age (years, mean ± SD)	75.8 ± 6.5	71.8 ± 5.3
Height (cm, mean ± SD)	156.8 ± 8.6	157.7 ± 9.7
Weight (kg, mean ± SD)	60.4 ± 10.8	61.3 ± 11.4
BMI (mean ± SD)	24.6 ± 4.0	24.6 ± 3.5
BMI > 25 (*n*, %)	15 (42.9%)	14 (40.0%)
Smoking (%)	3 (8.6%)	3 (8.6%)
Alcohol (%)	2 (5.7%)	2 (5.7%)
Medical history		
Hypertension (%)	17 (48.6%)	17 (48.6%)
Diabetes (%)	8 (22.9%)	7 (20.0%)
Heart disease (%)	8 (22.9%)	4 (11.4%)
Pulmonary disease (%)	2 (5.7%)	5 (14.3%)
Cerebrovascular disease (%)	2 (5.7%)	1 (2.9%)
Renal disease (%)	2 (5.7%)	3 (8.6%)
Mallampati score		
1	22 (62.9%)	19 (54.3%)
2	8 (22.9%)	10 (28.6%)
3	3 (8.6%)	4 (11.4%)
4	2 (5.7%)	2 (5.7%)
Midazolam (mg, mean ± SD)	2.7 ± 0.5	2.7 ± 0.8
Propofol (mg, mean ± SD)	14.6 ± 11.5	12.6 ± 12.7
Pethidine (mg, mean ± SD)	12.9 ± 22.2	11.4 ± 21.3

**Table 2 jcm-09-03282-t002:** Incidence of hypoxia in both groups.

	Pre-Oxygenated Group (*n* = 35)	Non-Oxygenated Group (*n* = 35)	*p* Value
Hypoxia before procedure (no., %)	0 (0%)	1 (2.9%)	0.314
SaO_2_ before procedure (mean ± SD)	95 ± 2	96 ± 2	0.726
Hypoxia during procedure (no., %)	0 (0%)	28 (80%)	<0.001

**Table 3 jcm-09-03282-t003:** Relative risk (95% CI) of hypoxia by group using bootstrapping in logistic regression model.

Outcome (Hypoxia)	Incidence Cases, *n* (%)	Crude OR (95% CI)
Pre-oxygenated group	28/35 (80.0)	1.00 (reference)
Non-oxygenated group	0/35 (0.0)	13.12 (12.96–13.29)

CI, confidence interval; OR, odds ratio.

**Table 4 jcm-09-03282-t004:** Sub-group analysis of non-oxygenated group (n = 35).

	Hypoxia Group (*n* = 28)	Non-Hypoxia Group (*n* = 7)	*p* Value
Sex (male, %)	14 (50.0%)	2 (28.6%)	0.415 *
Age (years, mean ± SD)	71.3 ± 5.4	73.9 ± 4.7	0.260
Height (cm, mean ± SD)	158.9 ± 9.8	153.1 ± 8.2	0.164
Weight (kg, mean ± SD)	62.5 ± 11.8	57.0 ± 8.8	0.266
BMI (mean ± SD)	24.6 ± 3.5	24.4 ± 3.7	0.867
BMI > 25 (*n*, %)	13 (46.4%)	1 (14.3%)	0.203 *
Smoking (%)	3 (10.7%)	0 (0%)	>0.999 *
Alcohol (%)	2 (7.1%)	0 (0%)	>0.999 *
Medical history			
Hypertension (%)	11 (39.3%)	6 (85.7%)	0.041 *
Diabetes (%)	7 (25.0%)	0 (0%)	0.301 *
Cardiovascular disease (%)	2 (7.1%)	2 (28.6%)	0.171 *
Pulmonary disease (%)	4 (14.3%)	1 (14.3%)	>0.999 *
Cerebrovascular disease (%)	1 (3.6%)	0 (0%)	>0.999 *
Renal disease (%)	3 (10.7%)	0 (0%)	>0.999 *
Mallampati score 1,2 vs. 3,4			0.311
1	15 (53.6%)	4 (57.1%)	
2	7 (25.0%)	3 (42.9%)	
3	4 (14.3%)	0 (0%)	
4	2 (7.1%)	0 (0%)	
Midazolam (mg, mean ± SD)	2.9 ± 0.6	2.1 ± 1.1	0.022
Propofol (mg, mean ± SD)	11.8 ± 11.2	15.7 ± 18.1	0.472
Pethidine (mg, mean ± SD)	14.3 ± 23.0	0	0.003

BMI, body mass index; SD, standard deviation. * Fisher’s exact test.

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
