# Peer review of "Oxygenation before Endoscopic Sedation Reduces the Hypoxic Event during Endoscopy in Elderly Patients: A Randomized Controlled Trial"

_jcm, 2020, doi:10.3390/jcm9103282_

Round 1

Reviewer 1 Report

The manuscript describes a prospective randomised controlled trial of oxygen therapy during sedation.

Comments for the authors

Page one, line 30. “… Remove patient anxiety and pain…” Is a bit ambitious. I think “… Decrease patient anxiety and pain…” Might be better.

You tell us that the patients received either propofol and midazolam. It’s not clear at this stage whether that was entirely at the choice of the sedationist? Were there any differences in characteristics between the midazolam patients or the propofol patients? What were the doses?

Table 1. Baseline characteristics. You have already told us that the patients were allocated to the 2 treatment groups at random. Therefore any difference between them can also only be due to random chance and there is no point in statistical testing of these so the column of P values should be removed from this table.

Superficially, your results are “obvious”. It is not a surprise to me that failure to supplement oxygen is associated with increased risk of hypoxaemia. Indeed it’s this factor that led various authors of guidelines (including myself) to clearly specify the benefit of supplementary oxygen. However I think there is more that you could be doing with your data.

Could you compare the patients receiving midazolam (and their characteristics) and those receiving propofol (and their characteristics)?  I would also be interested to see if there is any relationship between the dose of propofol and the amount of hypoxia)? You could address this by measuring the area under the curve of the oxygen saturation versus time plot for each individual patient and then perhaps by putting a graph of that ( percent  X minutes) versus drug dose (i.e. propofol mg or propofol mg/min).

Now that the matter of oxygen supplementation is settled (it is a good thing) I encourage you not to do any further sedation studies involving control groups without oxygen.

Author Response

Page one, line 30. “… Remove patient anxiety and pain…” Is a bit ambitious. I think “… Decrease patient anxiety and pain…” Might be better.

: I fully agree with you. I revised “Remove” to “Decrease”.

You tell us that the patients received either propofol and midazolam. It’s not clear at this stage whether that was entirely at the choice of the sedationist? Were there any differences in characteristics between the midazolam patients or the propofol patients? What were the doses?

: Most of our protocols of sedation are midazolam based balanced propofol sedation that initial 2-3 mg midazolam was given to patients followed by additional propofol 10 – 20 mg until patient was sedated enough. And in Korea, sedative endoscopy has been performed with NAAP manner (non-anesthesiologist administration of propofol) so, the choice of sedation is on endoscopist. Exceptionally, we adopted propofol mono sedation protocol as an alternative regimen for the patients with liver disease especially liver cirrhosis. In case of propofol sedation, we used 30mg propofol as initial dose and 10 – 20mg propofol as additional doses consecutively. I describe above contents on discussion - the limitation of study. As below.

“The third limitation was our sedation protocols. We adopted midazolam based balanced propofol sedation that initial 2-3 mg midazolam was given to patients followed by additional propofol 10 – 20 mg until patient was sedated enough. Exceptionally, we adopted propofol mono sedation protocol as an alternative regimen for the patients with liver disease especially liver cirrhosis. In case of propofol sedation, we used 30mg propofol as initial dose and 10 – 20mg propofol as additional doses consecutively. So, this effect of preoxygenation to sedation may concern mainly midazolam based balanced propofol sedation rather than propofol mono sedation. The effect of preoxygenation on propofol mono sedation, more study may be needed.”

Table 1. Baseline characteristics. You have already told us that the patients were allocated to the 2 treatment groups at random. Therefore any difference between them can also only be due to random chance and there is no point in statistical testing of these so the column of P values should be removed from this table.

: I fully agree with you. I removed P value on table 1.

Superficially, your results are “obvious”. It is not a surprise to me that failure to supplement oxygen is associated with increased risk of hypoxaemia. Indeed it’s this factor that led various authors of guidelines (including myself) to clearly specify the benefit of supplementary oxygen. However I think there is more that you could be doing with your data.

: Many endoscopist would know the benefits of supplementary oxygen during endoscopy. Nevertheless, on many guidelines, there is no recommendation how we perform the sedation during endoscopy more safely to the high risk patients especially old age patients. So, in this study, we emphasized pre-oxygenation should be applied specifically to old age patients. 

Could you compare the patients receiving midazolam (and their characteristics) and those receiving propofol (and their characteristics)?  I would also be interested to see if there is any relationship between the dose of propofol and the amount of hypoxia)? You could address this by measuring the area under the curve of the oxygen saturation versus time plot for each individual patient and then perhaps by putting a graph of that ( percent  X minutes) versus drug dose (i.e. propofol mg or propofol mg/min).

: I wonder too if propofol could be a key roll for hypoxia, however there was no relationship the dose of propofol and hypoxia on additional analysis. Total propofol doses of hypoxia group and non-hypoxia group were 11.8 ± 11.2mg, 14.8 ± 12.5mg respectively (p=0.315). When we divided total dose of propofol into 0~10mg, 20mg or more, there were no significant difference (p=0.488).

I guess the reason why the dose of propofol was not related with hypoxia could be that we used small doses of propofol because we adopted almost midazolam base protocol in contrast 2 cases of propofol mono protocol.

Now that the matter of oxygen supplementation is settled (it is a good thing) I encourage you not to do any further sedation studies involving control groups without oxygen.

: I will remind your kind advice on my further research. I so appreciate you.

Reviewer 2 Report

In this study, was the pre-oxygenation terminated when endosocpy started?

During sedated endoscopy, we usually start oxygenation when patients’ SpO2 level starts to fall. Do the authors think that the pre-oxygenation for all sedated endoscopy for elderly patients might replace the on-demand oxygenation?  If so, what is the advantage of pre-oxygenation over the on-demand method?

Author Response

During sedated endoscopy, we usually start oxygenation when patients’ SpO2 level starts to fall. Do the authors think that the pre-oxygenation for all sedated endoscopy for elderly patients might replace the on-demand oxygenation?  If so, what is the advantage of pre-oxygenation over the on-demand method?

: I fully agree with you. We also used oxygenation on demand manner until quite recently. Though almost of hypoxia during endoscopy without oxygenation can be easily improved, we have experienced a few cases of severe hypoxia needed airway maneuver or AMBU bagging. In this study, we did not meet the severe cases but, there were not hypoxic event even in elderly patients. Furthermore, in Korea, there have been many legal problems related sedative endoscopy. It is trend that more safe sedation rather than effective sedation is emphasized. So, I think that at least a recommendation would be needed for decrease of hypoxic risk in high risk patients.
